# An Evaluation of Nutritional Status and Problems with Dietary Compliance in Polish Patients with Celiac Disease

**DOI:** 10.3390/nu14132581

**Published:** 2022-06-22

**Authors:** Malgorzata Kostecka, Joanna Kostecka-Jarecka, Katarzyna Iłowiecka, Julianna Kostecka

**Affiliations:** 1Department of Chemistry, Faculty of Food Science and Biotechnology, University of Life Sciences, Akademicka 15, 20-950 Lublin, Poland; 2Independent Public Healthcare Center in Łęczna, Krasnystawska 52, 21-010 Leczna, Poland; kostecka-joanna@wp.pl; 3Department of Food and Nutrition, Medical University of Lublin, Chodźki 4a, 20-093 Lublin, Poland; katarzyna.ilowiecka@umlub.pl; 4Faculty of Medicine, Medical University of Lublin, Chodźki 19, 20-093 Lublin, Poland; kostecka.julianna@gmail.com

**Keywords:** celiac disease, gluten-free-diet, nutritional status, gluten-containing foods, diets

## Abstract

Celiac disease (CD, enteropathy) is a genetic autoimmune disease (abnormal immune response that attacks healthy tissues) associated with gluten intolerance. The aim of this study was to evaluate and monitor the nutritional status of CD patients, explore the problems associated with diet planning and dietary adherence among children and adults, and assess the impact of these factors on the persistence of CD symptoms. This study was carried out as part of the project entitled “A gluten-free diet without obstacles—eating well and healthy” (POWR 03.01.00-00-T153/18), conducted in Lublin Voivodeship. The study involved 87 persons, including 23 children younger than 18. At the beginning of the study and after nine months, all adult participants (older than 18) were subjected to a body composition analysis with the SECA mBCA 515 analyzer. During the project, the participants attended three consultations with a dietician. During each visit, the subjects’ body weight, nutritional status and diets were evaluated; their diets were modified, and problems relating to dietary adherence were resolved. The initial body composition analysis revealed a risk of sarcopenic obesity in 30% of adult participants, in particular in women (*p* = 0.003) older than 45 (*p* = 0.001). The risk of being underweight was diagnosed in 25% of the subjects, in particular, in women younger than 35 (*p* = 0.0023) and in participants who had been affected by short stature and underweight in childhood, i.e., before CD diagnosis (*p* = 0.0024). The analysis demonstrated that patients with gastrointestinal symptoms (abdominal pain, diarrhea, vomiting) of CD were significantly more likely to avoid even accidental exposure to gluten and were more likely to strictly follow GFD recommendations (1.97; 95CI:1.56–2.12, *p* = 0.0001) and safety guidelines when preparing meals at home (1.76; 95CI: 1.34–192, *p* = 0.0023). Parents, in particular, parents of toddlers and preschoolers who are at significantly higher risk of CD, adhered strictly to dietary guidelines and did not allow for any exceptions when preparing meals (1.88; 95CI: 1.53–2.09, *p* = 0.001). Persons at risk of malnutrition were also far less likely to deliberately choose gluten-containing foods (0.74; 95CI: 0.53–0.91, *p* = 0.021), in particular, patients with Marsh type 3a and 3b classification (*p* = 0.01) and persons whose intestinal histology scores did not fully improve after switching to a GFD. An assessment of the effectiveness of diet therapy based on the phase angle revealed that dietary recommendations had a positive impact on patients who had been recently diagnosed with CD. In all age groups, the main problem was accidental exposure to gluten, in particular in foods that were not labeled with the crossed grain symbol. A comparative analysis of CDAT questionnaires revealed that dietary advice on eating out significantly improved adherence to a GFD and reduced the frequency of unintentional gluten exposure in all age groups.

## 1. Introduction

Celiac disease (CD, enteropathy) is a genetic autoimmune disease (abnormal immune response that attacks healthy tissues) associated with gluten intolerance [1]. Until recently, CD had been regarded as a condition that affects mainly children aged 4 to 24 months, where symptoms appear after the introduction of gluten to the diet. At present, 60% of new cases are diagnosed in adults (disease incidence is highest in the 30–40 age group), including in 15–20% of adults older than 60 [2]. The diagnosis of CD can be challenging since symptoms can vary significantly from patient to patient [3].

A gluten-free diet (GFD) is the mainstay of the treatment of CD [4,5,6]. In small children, a GFD generally leads to an improvement in symptoms within several weeks. In adults, CD symptoms take longer to subside, and histological features improve only after 3–12 months. The aim of treatment is not only to eliminate symptoms, regenerate intestinal villi, and improve the patient’s nutritional status, but also to minimize the risk of cancer. Patients diagnosed with CD have to adhere to a GFD for the rest of their lives, which can negatively affect the quality of their lives, lead to psychological problems, fear of accidental gluten exposure [7], vitamin and mineral deficiency, metabolic disorders, increased risk of cardiovascular disorders, and chronic constipation [8,9]. Most disadvantages of GFD can be eliminated or minimized by informing the patient about the risks associated with failure to adhere to the diet, seeking help from an experienced dietician, providing support in planning meals, and making informed food choices. It should be noted that Polish celiac patients are not reimbursed for GFD products. In March 2019, the Polish Celiac Society filed a petition (signed by 12,400 citizens) to the Polish Ministry of Health demanding that GFD products be included in the reimbursement scheme. The Ministry responded by stating that: “At present, the Ministry of Health is not conducting legislative works aimed at refunding gluten-free products” [10]. Access to a registered dietitian experienced in CD is also limited in Poland, and consultation, when available, focuses primarily on the elimination of gluten from the diet [11]. Patients diagnosed with CD receive brochures with basic information about GFD. The brochures have been developed by the Polish Celiac Society, the main nationwide organization that supports people on a GFD. Unfortunately, the Society does not organize regular meetings with dietitians or diet therapies for individual patients. Most CD patients in Poland attend a single consultation with a private dietitian. Then they learn how to compose gluten-free meals on their own, and search for support in local groups or social media, which can lead to mistakes in GFD compliance. A GFD should not be implemented based only on a single consultation that focuses on potential sources of gluten. According to the guidelines of the European Society for the Study of Celiac Disease [6], patients should learn not only to eliminate gluten from their diets, but should be also made aware that a GFD, like any diet, must be balanced—which can be a problematic issue, especially in institutional feeding [12]. Knowledge about the problems faced by patients adhering to a GFD is needed to adapt dietary interventions to individual needs, and it increases the effectiveness of educational measures to guarantee that patients do not make mistakes and better adhere to a GFD for positive health outcomes. In the Polish literature, there is a general scarcity of research examining GFD compliance in children and adults or the influence of dietary mistakes on the patients’ nutritional status and wellbeing.

Therefore, the aim of this study was to evaluate and monitor the nutritional status of CD patients, explore the problems associated with diet planning and dietary adherence among children and adults, and assess the impact of these factors on the persistence of CD symptoms.

## 2. Materials and Methods

### 2.1. Study Design and Participants

This study was carried out as part of the project entitled “A gluten-free diet without obstacles—eating well and healthy” (POWR 03.01.00-00-T153/18), conducted in Lublin Voivodeship between March 2019 and December 2021. The study was conducted at the Nutrition Clinic of the University of Life Sciences in Lublin, Poland, under the auspices of the Polish Celiac Society.

The study involved 87 persons, including 23 children younger than 18. The inclusion criteria were diagnosed enteropathy and treatment in a gastroenterology clinic. The patients were diagnosed based on the guidelines formulated by ESPGHAN and the Polish Society for Pediatric Gastroenterology, Hepatology and Nutrition [13,14,15,16,17], a positive result of the tissue transglutaminase (IgA and/or IgG) antibody test, and a positive biopsy result based on Marsh classification. The presence of comorbidities, including other autoimmune disorders, was not an exclusion criterion. The recruited subjects had been also diagnosed with vitiligo, rheumatoid arthritis, autoimmune thyroiditis, and type 1 diabetes (Table 1). Children and young adults who were diagnosed with celiac disease as well as type 1 diabetes did not modify their diets accordingly. The patients regularly consulted a diabetologist to improve glycemic control. All patients received insulin and introduced low carbohydrate alternatives to their diets.

### 2.2. Nutritional Status and Body Mass Composition

At the beginning of the study and after 9 months, all adult participants (older than 18) were subjected to a body composition analysis with the SECA mBCA 515 analyzer, and the following parameters were measured: BMI, fat mass, fat-free mass, skeletal bone mass, abdominal adipose tissue, total body water, and phase angle. A bioelectrical impedance vector analysis (BIVA) was performed. The results were used to evaluate the subjects’ body composition and the observed changes. During consultations with a dietician, the participants presented 3-day food diaries (developed based on photographs of foods and meals [18] and completed a questionnaire composed of 36 questions. The collected information was used to analyze the subjects’ nutritional status and dietary habits, and the results will be presented in an upcoming study. Children (younger than 18) had their height and body weight measured on the SECA 799 electronic column scales, and the results were evaluated with the use of growth charts for the Polish population [19]. The children’s nutritional status and somatic development were assessed.

### 2.3. Dietary Consultations

During the project, the participants attended three consultations with a dietician. During each visit, the subjects’ body weight, nutritional status, and diets were evaluated; their diets were modified, and problems relating to dietary adherence were resolved. Dietary adherence was assessed with the use of two validated questionnaires: (i) a GFD compliance questionnaire [20] and (ii) the Celiac Disease Adherence Test (CDAT) [21]. For pediatric patients, the questionnaires were filled by the parents or guardians, whereas older children actively participated in the process, which is common practice in research studies involving dietary recalls [22,23,24]. The results were used to assess the patients’ attitudes to a GFD and discuss problems that could undermine dietary adherence.

### 2.4. Survey Questionnaire

The questionnaire was composed of three parts. In the first part, the participants provided information about their medical history, CD diagnosis, symptoms before CD diagnosis and after GFD implementation, results of serological tests, biopsy results based on Marsh classification, comorbidities, including autoimmune disorders, gluten-dependent disorders in the family, time since GFD implementation, and changes in health status after gluten elimination. In the second part of the questionnaire, the participants described problems with dietary compliance, meal preparation techniques, safety standards in preparing gluten-free meals, and knowledge about recommended foods, unsafe foods, and gluten-free food labels. In the third part of the questionnaires, the subjects were asked to describe their lifestyle and physical activity, list medications and supplements they were taking, and indicate their gender and age. The nutritional survey involved an analysis of 3-day food diaries to assess the participants’ food choices and determine whether their macronutrient, vitamin, and mineral requirements were adequately met.

### 2.5. Data Analysis

Categorical variables were presented as sample percentages (%), and continuous variables were expressed by median values. The differences between groups were analyzed in the chi-squared test (categorical variables) or the Mann-Whitney test (continuous variables). Before statistical analysis, data were checked for normal distribution in the Kolmogorov-Smirnov test. The odds ratios (ORs) and 95% confidence intervals (95% CIs) were calculated. The reference categories (OR = 1.00) included adherence to persons with a healthy nutritional status based on phase angle values or persons with healthy BMI, and persons with other symptoms of celiac disease. The ORs were adjusted for adherence- to a gluten-free diet when eating out, consumption of foods containing gluten, and accidental exposure to gluten. The significance of ORs was assessed by Wald’s statistics. The results of all tests were regarded as statistically significant at *p* < 0.05. Data were processed in the Statistica program (version 13.1 PL; StatSoft Inc., Tulsa, OK, USA; StatSoft, Krakow, Poland).

## 3. Results

The study involved 87 persons (including 14 males, 16.1%). Celiac disease was diagnosed at the mean age of 6.7 ± 3.2 years in children and 31.5 ± 11.9 years in adults. The time between symptom onset and diagnosis was shorter in children (four months on average) than in adults (11 months on average) (*p* = 0.001). Based on the data obtained from patient interviews, only 10 CD patients had consulted a dietician, and 62% of the participants had rigorously adhered to a GFD.

The main symptoms before diagnosis involved gastrointestinal problems, dermatological problems, and growth disorders in children (Table 2). In most patients, these symptoms were eliminated or minimized after the implementation of a GFD. In 60% of the studied children, the transition to a GFD contributed to an increase in height and weight. In four children, growth disorders were additionally associated with a growth hormone deficiency, and the remaining children were between the 10th and 75th percentile and did not have growth disorders. In adults with short stature, the transition to a GFD had no apparent benefits because their growth had already been completed. In subjects with neurological comorbidities, the CD diagnosis and elimination of gluten from the diets had no effect on these disorders, and the transition to a GFD decreased the frequency and severity of epileptic episodes in only two cases.

In most participants, the transition to a GFD improved intestinal health, decreased the frequency of, and, consequently, eliminated abdominal pain and diarrhea. Anemia was eliminated and vitamin B_12_ absorption improved in 60% of the subjects. The transition to a GFD decreased the severity of dermatological problems in 90% of the patients. According to 56% of the surveyed population, their health had improved after switching to a GFD (*p* = 0.01), and 74% of the subjects reported an improvement or a considerable improvement in mood and wellbeing (*p* = 0.021).

### 3.1. Assessment of the Nutritional Status of Children

The mean age of children at the beginning of the study was 9.3 years (minimum—5 years and 4 months; maximum—16 years and 9 months). Children’s body weight varied considerably from 14.3 kg to 45.6 kg, and the mean body weight was 22.7 ± 11.2 kg. Height was determined in the range of 97.5 cm a 152.9 cm, and the mean height was 121.6 ± 15.9 cm. A growth evaluation based on the BMI percentile chart revealed that only six children (20.1%) were within the narrow gender- and age-based norm (25th to 75th percentile on the WHO growth chart). At the beginning of the study, the BMI values of nine children (39.1%) were below the 5th percentile line, which is indicative of short stature and underweight. The BMI values of the remaining children ranked between the 5th and 25th percentile for their age and gender. None of the children exceeded the 85th percentile for BMI at the onset of the study. After nine months, height and body weight increased in all children by 5.4 cm and 2.4 kg, respectively, on average (4.8 cm and 1.9 kg, respectively, in girls; 6.1 cm and 2.7 kg, respectively, in boys). The BMI values were below the fifth percentile in only five children (21.7%); rapid compensatory growth was noted in the remaining children, and the BMI values of three children exceeded the 85th percentile.

### 3.2. Body Composition Analysis and Evaluation of Nutritional Status in Adults

The initial body composition analysis revealed a risk of sarcopenic obesity in 30% of adult participants, in particular in women (*p* = 0.003) older than 45 (*p* = 0.001). The risk of being underweight was diagnosed in 25% of the subjects, in particular in women younger than 35 (*p* = 0.0023) and in participants who had been affected by short stature and underweight in childhood, i.e., before CD diagnosis (*p* = 0.0024). Muscle mass increased in only seven subjects, all of them male and younger than 45 (*p* = 0.003). The bioimpedance vector analysis (BIVA) revealed a decrease in body cell mass and a higher risk of wasting in 11 patients, and it confirmed that the deterioration of nutritional status was associated with a decrease (underweight) and increase (obesity) in hydration. A body composition analysis performed after nine months revealed a lower risk of wasting (only two subjects) and a higher number of patients with increasing muscle mass. This result was noted in 23 patients, including male subjects younger than 45 and women aged 18–30 (*p* = 0.001), where the increase in muscle mass was significantly correlated with a decrease in the fat mass index (FMI, by 1.6 kg/m^2^ on average) (*p* = 0.001). The BIVA also revealed that an increase in muscle mass was correlated with an increase in body cell mass in women. The results of the body composition analysis are presented in Table 3.

The phase angle is an important parameter in nutritional status assessments. A higher phase angle is indicative of improving cell membrane health. In the studied population, the phase angle ranged from very low values (3.5–4.0°) in subjects at risk of sarcopenic obesity (*p* = 0.001) to 5–6° in more than 2/3 of the participants. A phase angle is a valuable tool for evaluating the effectiveness of diet therapy, and this parameter was measured at the beginning and after nine months of the study. The results revealed an improvement in the nutritional status of high-risk CD patients (recently diagnosed subjects who adhered to a GFD for two to four weeks and reported an improvement in the function of intestinal villi and nutrient absorption). In these subjects, the phase angle increased by 0.9 ± 0.2° on average (*p* = 0.0021), whereas in the entire studied population, the phase angle increased by 0.4 ± 0.3°.

### 3.3. Evaluation of Adherence to a Gluten-Free Diet

The extent to which the participants adhered to a GFD was evaluated with two validated questionnaires. The survey was conducted during the first consultation with a dietician, and it involved an assessment of the patients’ compliance with dietary recommendations after diagnosis and after switching to a GFD. The second evaluation was performed nine months after the first consultation to determine the patients’ adherence to dietary recommendations and modification of previous eating habits (Table 4).

At the beginning of the study, adults scored 19.4 ± 7.8 points on average, which is indicative of low adherence to a GFD, whereas the average score among children was 15.8 ± 10.3, which points to average compliance with GFD guidelines. Regardless of age, the main problem was accidental exposure to gluten, in particular in processed foods that were not labeled as gluten-free. Many adults deliberately selected foods containing gluten or trace amounts of gluten when eating out. This behavior was more frequently reported by adults who did not experience gastrointestinal symptoms after eating gluten (*p* = 0.002) and children aged 12–14 who were embarrassed to follow GFD recommendations during interactions with peers. Gluten-containing foods were consumed 3–5 times on average in the previous four weeks, but 17% of the participants deliberately selected gluten-based foods 6–10 times on average in the previous four weeks, whereas 5.3% consumed gluten more than 10 times in the evaluated period (*p* < 0.05). The failure to comply with GFD guidelines and the consequences of regular exposure to gluten were discussed in detail during consultations with a dietician. A survey of the participants’ eating habits and dietary adherence conducted nine months after the first consultation revealed an improvement in the patients’ nutritional awareness and greater adherence to a GFD. Adults scored 14.1 ± 6.3 points on average, which is indicative of satisfactory dietary adherence, whereas children scored 10.9 ± 4.6 points on average, which is indicative of highly satisfactory dietary adherence. The frequency of deliberate exposure to gluten decreased significantly in both groups (*p* < 0.05), and only 12.8% of adults consumed gluten deliberately 3–5 times in the previous four weeks, whereas the remaining adults did not eat gluten-containing foods or consumed gluten not more than 1–2 times in the analyzed period. Children consumed gluten 1–2 times in the previous four weeks, and gluten-containing foods were still more frequently selected by teenagers (14–16) (*p* = 0.003), 3–5 times on average in the studied period. Younger children who do not make their own dietary choices did not seek gluten-containing foods. A comparison of CDAT questionnaires revealed that dietary consultations during which the participants learned how to follow a GFD when eating out significantly improved dietary adherence (*p* = 0.0031) in all age groups. Teenagers (14–16), in particular girls (*p* = 0.004), were the only group where nutrition education did not improve dietary adherence, especially during interactions with peers.

The participants’ adherence to a GFD was also analyzed with the use of a questionnaire which revealed that 1/3 of adults did not observe dietary recommendations, mostly by consuming (1–2 times per month) small servings of foods or meals containing gluten. Failure to adhere to dietary guidelines was more frequently observed among persons without gastrointestinal symptoms (*p* = 0.001), patients not diagnosed with Duhring’s disease (*p* = 0.002), and patients who had followed a GFD for more than one year (*p* = 0.034). Most notably, when eating out, 25.6% of adults did not inform caterers that they were sensitive to gluten. The main reasons for the above were a perceived lack of understanding, including from waiters (*p* = 0.002), a low number of venues serving gluten-free meals (*p* = 0.005), and in 30% of the cases—embarrassment and fear of criticism. In the group of the studied children, 90% of the parents made attempts to order gluten-free meals when eating out or did not take their children out to food venues (*p* = 0.002). More than 90% of the participants read food labels after switching to a GFD, whereas only 54% of the patients read food labels 9 months after the dietary consultation. The majority of the respondents shopped for food “from memory” (*p* = 0.004), selected products with a crossed grain symbol (*p* = 0.001), or learned to buy gluten-free foods without reading the list of ingredients on the label (*p* = 0.004). The crossed grain symbol was important for 3/4 of the participants, but patients who adhered to a GFD for a long period of time tended to incorporate more naturally gluten-free products and foods labeled as gluten-free into their diets, and the crossed grain symbol did not play a decisive role in their food choices. All of the surveyed children were familiar with and understood the meaning of the crossed grain symbol on food labels.

### 3.4. Correlation between Nutritional Status and Adherence to a Gluten-Free Diet among Adults

An analysis of the dietary behaviors of adults adhering to a GFD revealed a correlation between their nutritional status, degree of dietary compliance, and frequency with which they deliberately selected gluten-containing foods. The obtained results were presented in Table 5.

The analysis demonstrated that patients with gastrointestinal symptoms (abdominal pain, diarrhea, vomiting) of CD were significantly more likely to avoid even accidental exposure to gluten and were more likely to strictly follow GFD recommendations (1.97; 95CI:1.56–2.12, *p* = 0.0001) and safety guidelines when preparing meals at home (1.76; 95CI: 1.34–192, *p* = 0.0023). Parents, in particular parents of toddlers and preschoolers who are at significantly higher risk of CD, adhered strictly to dietary guidelines and did not allow for any exceptions when preparing meals (1.88; 95CI: 1.53–2.09, *p* = 0.001). Persons at risk of malnutrition were also far less likely to deliberately choose gluten-containing foods (0.74; 95CI: 0.53–0.91, *p* = 0.021), in particular, patients with Marsh type 3a and 3b classification (*p* = 0.01) and persons whose intestinal histology scores did not fully improve after switching to a GFD. In turn, subjects with BMI > 25 kg/m^2^ were significantly more likely to make minor exceptions to dietary recommendations and purchase food products that, according to label information, could contain trace amounts of gluten (*p* = 0.031).

## 4. Discussion

A GFD is the only available treatment for persons diagnosed with CD. Patients who follow a diet therapy, adhere to GFD guidelines, observe the recommended meal preparation techniques, and learn to cook both gluten-based and gluten-free meals at home are able to eliminate the most problematic symptoms, regenerate intestinal villi, and improve nutrient absorption within just four to six weeks. All of the above factors influence the patients’ nutritional status.

Most persons with CD do not seek expert advice when planning their diets. As a result, not all patients make appropriate food choices or apply correct meal preparation techniques. In the present study, only 10 patients diagnosed with CD consulted a dietician, and 62% of the participants rigorously adhered to a GFD. Less satisfactory results were reported in India, where only 53% of the participants complied with dietary guidelines [18], and in Greece, where only 58% of children with CD adhered to a GFD [19]. In turn, in a Canadian study of children with CD, 95% of the respondents strictly observed nutritional recommendations [25]. Strict adherence to a GFD may be more challenging in children and adolescents than in adults [26]. According to the North American Society of Pediatric Gastroenterology, Hepatology, and Nutrition, compliance with GFD ranges from 45% to 81% in children [27]. In the current study, adherence to a GFD was higher among children than adults.

Failure to follow a GFD poses a serious health problem and the greatest challenge for physicians and dieticians. In the present study, adults with CD did not observe dietary recommendations due to the higher cost, lower availability, and lower selection of gluten-free products than conventional foods, the need to eat out, and the fear that their dietary habits will not be approved by family and friends. In children, the main reasons for not adhering to a GFD were reluctance to eat out, peer pressure, and low preference for the taste of gluten-free foods, in particular bread, which is consistent with the results of the Indian study [6]. Peer pressure is an important factor that can cause teenagers to rebel against a restrictive diet [28,29]. In the present study, teenagers who frequently ate ready-to-eat foods claimed that these products were not clearly labeled. Similar observations were made by other authors [6,24,30].

Previous research has shown that patients experiencing a wide range of CD symptoms, mostly digestive symptoms, tended to adhere more strictly to a GFD, even without the support of a dietitian or a doctor. Tovili et al. observed that patients with a high risk of complications were more likely to comply with GFD guidelines than patients without classic CD symptoms [31]. These observations indicate that severe symptoms of CD and unresolved health issues can prompt patients to adhere to a GFD despite the encountered problems. Currently, the suggested protocols call for a first examination six months after the beginning of a GFD and every 18–24 thereafter, regardless of the patient’s clinical characteristics at diagnosis (“one size fits all” model) [6,14,32,33,34]. Unfortunately, in other countries as well as in Poland, the evaluation of a patient’s compliance with medical and dietary recommendations can be difficult. A patient is initially diagnosed in a gastroenterological clinic (most often in a clinical center), and further observations are generally conducted by several doctors (including GPs) from various centers. In addition, the percentage of patients who attend consultations with dieticians does not exceed 10%. As a result, the acquisition of reliable data and monitoring of GFD compliance can be challenging [35,36,37].

Malabsorption and malnutrition caused by damage to the intestinal mucosa can delay growth and cause short stature in children with a late CD diagnosis. In a study of the Finnish population, Nurminen et al. found that the severity of intestinal damage based on the Marsh classification can stilt growth in children diagnosed with CD [38]. In the present study, children with CD were significantly shorter than their peers, and the time of diagnosis was directly proportional to growth disorders. Other researchers confirmed that an early CD diagnosis decreases the risk of permanent growth disorders in children [39,40,41]. Weiss et al. [42] demonstrated that the growth rate of CD patients was inversely correlated with age at diagnosis. In turn, a Turkish study reported an inverse correlation between age at diagnosis and an increase in the weight and height of children with CD [43]. A GFD promotes gut mucosal healing and improves nutrient absorption. In the current study, catch-up growth was higher in all children and teenagers following a GFD. Other studies demonstrated that compensatory growth proceeds at a slower rate when CD is diagnosed in children older than four years than in infants or toddlers [39,44,45]. The rate of mucosal healing and catch-up growth varies across individuals. In other studies, faster growth was reported after six months of adherence to a GFD, and it continued for two to three years [44,46]. In the work of Boersman et al., the z-score was around 1 SD lower in children diagnosed with CD past the age of three years than in children who were diagnosed at an earlier age [46].

A gluten-free diet (GFD) is the cornerstone of CD treatment and management. At present, a GFD is also recommended for patients with other autoimmune conditions, such as psoriasis, multiple sclerosis (MS), type 1 diabetes (T1D), and autoimmune thyroid diseases (ATDs) [47,48,49]. The role of gluten in the immune response and its ability to trigger symptoms in patients with autoimmune diseases has attracted considerable research interest in recent years [47]. Autoimmune comorbidities in CD patients were also noted in this study. The failure to observe CFD guidelines can trigger or intensify the symptoms of CD and contribute to secondary autoimmunity [50].

Concentration problems, attention disorders, and aggression during interactions with peers are frequently reported by parents and adults with CD, in particular before the implementation of a GFD, which additionally discourages patients to comply with dietary guidelines [51]. In the present study, 61% of children had experienced cognitive and behavioral disorders, and 1/3 of the adult population reported symptoms of depression or mood disorders. Nervous system complications, perception disorders, and psychiatric problems such as attention deficit have been reported in 6% to 11% of patients with CD [52,53], but the pathophysiology of neurological and psychiatric disorders in the progression of CD remains unclear. The prevalence of behavioral disorders in patients with untreated CD is estimated at 21% [54], and depression [55,56] and personality disorders are most frequently reported in the adult population. According to the surveyed participants, the transition to a GFD led to an improvement in mood, alleviated symptoms of CD, minimized behavioral problems, and improved academic performance in children and teenagers [57].

### Strengths and Limitations

The study has several strengths. The studied population was diverse, and it involved patients who had been recently diagnosed with CD, as well as persons who had transitioned to a GFD several years earlier. The CDAT questionnaire supported an assessment of dietary adherence in patients who had acquired some knowledge on the subject and persons who were only beginning to eliminate gluten from their diets. In all cases, dietary consultations and long-term education led to an improvement in eating habits, meal planning, and adherence to a GFD. Adults were subjected to a body composition analysis, and the nutritional status of adults following a GFD was monitored.

The study also had limitations. Firstly, the nutritional status of children was assessed based only on anthropometric measurements and growth charts because a body composition analyzer designed specifically for children and adolescents was not available. Secondly, the participants were recruited from only one Polish voivodeship; therefore, even though the sample was diverse, it was relatively small. Additionally, the presented data were obtained from patient interviews or their medical history. Our study focused solely on dietary habits and the patients’ nutritional status. Lastly, other autoimmune comorbidities in the studied population could have influenced dietary restrictions and the patients’ nutritional status.

## 5. Conclusions

The implementation of a GFD eliminated or minimized symptoms of CD in most patients.

The dietary intervention and consultations with a dietician improved growth parameters in 80% of children with CD.

An assessment of the effectiveness of diet therapy based on the phase angle revealed that dietary recommendations had a positive impact on patients who had been recently diagnosed with CD.

In all age groups, the main problem was accidental exposure to gluten, in particular in foods that were not labeled with the crossed grain symbol.

A comparative analysis of CDAT questionnaires revealed that dietary advice on eating out significantly improved adherence to a GFD and reduced the frequency of unintentional gluten exposure in all age groups.

Patients who had adhered to a GFD for a long period of time tended to incorporate more naturally gluten-free products into their diet, and the crossed grain symbol on the label was not a decisive factor when shopping for food.

Persons affected by a wide range of CD symptoms with a predominance of digestive symptoms tended to adhere more strictly to a GFD.

Dietary consultations should be an integral part of CD treatment.

## Figures and Tables

**Table 1 nutrients-14-02581-t001:** Comorbidities with celiac disease.

	Adults *n* (%)	Children *n* (%)
Celiac disease	64 (100)	23 (100)
including:		
Duhring’s disease	11 (17.2)	7 (30.4)
Rheumatoid arthritis	9 (14.1)	1 (4.3)
Type 1 diabetes	3 (4.7)	6 (26.1)
Crohn’s disease	7 (10.9)	2 (8.6)
Autoimmune thyroiditis	12 (18.8)	0 (0.0)
Psoriasis	6 (9.4)	0 (0.0)
Vitiligo	3 (4.7)	0 (0.0)

**Table 2 nutrients-14-02581-t002:** Frequency of the main symptoms before and after diagnosis celiac disease.

	Symptoms before Diagnosis		Symptoms after the Implementation of a Gluten-Free			
	Adults *n* (%)	Children *n* (%)	Adults *n* (%)	*p*-Value	Children *n* (%)	*p*-Value
Abdominal pain and bloating	62 (96.8)	16 (69.5)	16 (25.0)	<0.05	7 (30.4)	<0.05
Diarrhea and steatorrhea	53 (82.8)	11 (47.8)	9 (14.1)	<0.05	4 (17.4)	<0.05
Constipation	5 (7.8)	6 (26.1)	1 (1.6)	<0.05	3 (13.1)	<0.05
Weight loss	7 (10.9)	7 (30.4)	1 (1.6)	<0.05	0	<0.05
Low height	9 (14.1)	15 (65.2)	9 (14.1)	ns	6 (26.1)	<0.05
Aphthous mouth ulcers	37 (57.8)	14 (60.8)	7 (10.9)	<0.05	10 (43.5)	<0.05
Enamel hypoplasia	6 (9.3)	11 (47.8)	6 (9.3)	ns	11 (47.8)	ns
Chronic fatigue	37 (57.8)	12 (52.2)	6 (9.3)	<0.05	1 (4.3)	<0.05
Concentration problems	39 (60.9)	19 (82.6)	4 (6.3)	<0.05	6 (26.1)	<0.05
Neurological disorders (peripheral neuropathy, ataxia, epilepsy)	6 (9.3)	3 (13.0)	6 (9.3)	ns	3 (13.1)	ns
Chronic headache, migraine	28 (43.7)	9 (39.1)	13 (20.3)	<0.05	3 (13.1)	<0.05
Joint pain	21 (32.8)	7 (30.4)	5 (7.8)	<0.05	2 (8.6)	<0.05
Brain fog	14 (21.8)	5 (21.7)	8 (12.5)	<0.05	2 (8.6)	<0.05
Dermatological issues (Duhring’s disease)	26 (40.6)	7 (30.4)	3 (4.7)	<0.05	5 (21.7)	<0.05
Fertility problems	12 (18.8)	0	9 (14.1)	<0.05	0	ns
Malabsorption of vitamin B12	13 (20.3)	4 (17.4)	5 (7.8)	<0.05	1 (4.3)	<0.05
Chronic anemia	20 (31.2)	9 (39.1)	6 (9.3)	<0.05	5 (21.7)	<0.05
Other	14 (21.8)	7 (30.4)	4 (6.3)	<0.05	2 (8.6)	<0.05

**Table 3 nutrients-14-02581-t003:** Initial body composition and changes after 9 months among adults with celiac disease.

	Initial Body Composition Analysis	Body Composition Analysis after 9 Months	*p*-Value
BMI (kg/m^2^)	24.7 ± 2.9	24.2 ± 3.5	ns
Body weight (kg)	67.4 ± 17.9	64.1 ± 18.5	0.031
Fat mass (kg)	23.75 ± 4.6	21.09 ± 5.2	0.002
Fat mass index (FMI)	9.1 ± 3.7	8.2 ± 3.4	0.0035
Fat-free mass (FFM) (kg)	40.82 ± 9.3	43.17 ± 8.6	0.0041
Skeletal muscle mass (SMM) (kg)	17.31 ± 8.3	18.7 ± 8.7	0.044
Total body water (TBW) (%)	46.3 ± 5.1	49.7 ± 3.8	0.002
Extracellular water (ECW) (%)	20.7 ± 1.8	21.5 ± 2.2	ns

**Table 4 nutrients-14-02581-t004:** Celiac Disease Adherence Test.

	After Switching to a Gluten-Free Diet (Mean Number of Points ± SD)		9 Months after Consultation with a Dietician		*p*-Value
	Adults	Children <18 Years	Adults	Children <18 Years	
How often did you experience low energy levels in the past 4 weeks?	2.9 ± 1.6	3.4 ± 1.2	2.4 ± 1.7	2.1 ± 1.4	0.002
How often did you experience headaches in the past 4 weeks?	2.1 ± 1.2	1.1 ± 1.6	1.1 ± 1.5	1,1 ± 1.4	0.034
Do you strictly adhere to a GFD when eating out?	3.3 ± 1.7	1.9 ± 1.6	2.4 ± 1.7	1.4 ± 1.1	0.001
Do you consider the health consequences of your food choices?	2.6 ± 1.8	1.5 ± 1.1	2.4 ± 1.0	1.2 ± 1.0	0.062
What is the significance of accidental gluten exposure for your health?	2.6 ± 1.6	1.4 ± 1.1	1.5 ± 1.3	1.2 ± 1.1	0.0013
How often did you deliberately consume gluten-containing foods in the past 4 weeks?	3.7 ± 1.5	2.8 ± 1.3	2.4 ± 1.3	1.5 ± 1.2	0.0004

**Table 5 nutrients-14-02581-t005:** Odds ratios (95% confidence interval) in an analysis of the relationships between nutritional status and adherence to a gluten-free diet among adults.

	Persons at Risk of Malnutrition (ref. Persons with a Healthy Nutritional Status Based on Phase Angle Values)	Persons with BMI > 25 [kg/m^2^] (ref. Persons with Healthy BMI)	Persons with Gastrointestinal Symptoms (ref. Persons with Other Symptoms of Celiac Disease)
Strict adherence to a gluten-free diet when eating out	1.56 ** (1.24–1.88)	0.76 * (0.54–1.01)	1.97 ** (1.56–2.12)
Deliberate consumption of foods and meals containing gluten	0.74 * (0.53–0.91)	1.03 (0.98–1.21)	0.66 ** (0.51–0.84)
Accidental exposure to gluten	0.94 (0.78–1.06)	1.47 * (1.17–1.79)	0.75 * (0.58–0.93)

*—*p*-Value < 0.05; **—*p*-Value < 0.01

## Data Availability

Due to ethical restrictions and participant confidentiality, data cannot be made publicly available.

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
