# Peer review of "An Evaluation of Nutritional Status and Problems with Dietary Compliance in Polish Patients with Celiac Disease"

_nutrients, 2022, doi:10.3390/nu14132581_

Round 1

Reviewer 1 Report

The paper entitled “An evaluation of nutritional status and problems with dietary 2 compliance in patients with celiac disease” analyses GFD compliance, and its improvement after dietitian consultation, in a small sample of adults and children with celiac disease. It offers some interesting conclusions. However, there are relevant points to be improved and clarified.

-       In the children/adolescents, did you find any difference in GFD compliance according to having or not a biopsy-sparing diagnosis? Please comment on this point and add the % of subjects with or without a biopsy-free CD diagnosis.

-       Celiac patients also having T1 diabetes have to comply with additional dietary advice: did you find any difference in them regarding GFD compliance? Please add.

-       How many centers were involved?

-       Which GFD compliance questionnaire was used? Was it a validated one? Please add a reference. And also add a reference for CDAT questionnaire

-       How was GFD compliance assessed in children? An age-specific adapted questionnaire should be used.

-       Don’t you think that unbalanced gender distribution could have affected your results? Please comment about this eventuality.

-       It would be more informative if symptoms in table 2 are presented spit for children and adults

-       You reported that: “the transition to a GFD decreased the frequency and severity 161 of epileptic episodes in only two cases.”. Please, discuss this result: was cerebral MR/CT performed in these patients? Were there cerebral calcifications?

-       “In most participants, the transition to a GFD improved intestinal health and contributed to the regeneration of intestinal villi, which reduced the frequency of and, subsequently, eliminated abdominal pain and diarrhea. Hematological parameters (hemoglobin, hematocrit, red blood cell counts, iron, ferritin) improved.” These sentences are not supported by numerical data.

-       What about BMI centiles (or z-scores) in children? Please add these data.

-       “These observations indicate that severe symptoms of CD and unresolved health issues can prompt patients to adhere to a GFD despite the encountered problems.” This is not a result and should be moved to the discussion section.

-       Some of the data commented on in the discussion are not reported in the results (i.e. only 10 patients diagnosed with CD consulted a dietician)

-       Is there a refunding policy for GFD products for celiac patients in your country? Please add this point, because it could affect GFD compliance and vary among different countries.

Author Response

Responses to Reviewer #1

(changes in the manuscript are marked in green)

Dear Reviewer,

We are grateful for your critical assessment which has enabled us to improve the quality of our manuscript. Our point-by-point responses to your comments are presented below. The manuscript was revised in the “track changes” mode. We hope that the revised manuscript is suitable for publication in Nutrients.

“The paper entitled “An evaluation of nutritional status and problems with dietary 2 compliance in patients with celiac disease” analyses GFD compliance, and its improvement after dietitian consultation, in a small sample of adults and children with celiac disease. It offers some interesting conclusions. However, there are relevant points to be improved and clarified.”

Point 1: In the children/adolescents, did you find any difference in GFD compliance according to having or not a biopsy-sparing diagnosis? Please comment on this point and add the % of subjects with or without a biopsy-free CD diagnosis.

Response 1: We are grateful to the Reviewer for raising this important issue. According to the standard CD diagnostic procedure in Poland, the biopsy-sparing approach and Marsh classification are the mainstays for diagnosing the disease in both children and adults. Gastroscopies and biopsies had been performed in all study participants (including children), and these procedures had been carried out several times in around 10% of the patients. Therefore, compliance with GFD guidelines was not influenced by the diagnosis. However, we emphasized that patients with gastrointestinal symptoms of CD were significantly more likely to strictly follow GFD recommendations. The relevant information was given in the manuscript: “The patients were diagnosed based on the guidelines formulated by ESPGHAN and the Polish Society for Pediatric Gastroenterology, Hepatology and Nutrition [11-15], a positive result of the tissue transglutaminase (IgA and/or IgG) antibody test, and a positive biopsy result based on Marsh classification.” (lines 103-106). We hope that this information answers your query.

Point 2: Celiac patients also having T1 diabetes have to comply with additional dietary advice: did you find any difference in them regarding GFD compliance? Please add.

Response 2: Thank you for this important question. To answer your query, the following sentences were added in the Materials and Methods section of the revised manuscript: “Children and young adults who were diagnosed with celiac disease as well as type 1 diabetes did not modify their diets accordingly. The patients regularly consulted a diabetologist to improve glycemic control. All patients received insulin and introduced low carbohydrate alternatives to their diets.”     (lines 109-112)

Point 3: How many centers were involved?

Response 3: Thank you for raising this issue. The study was conducted at the Nutrition Clinic of the University of Life Sciences in Lublin, Poland, under the auspices of the Polish Celiac Society. The relevant information was provided in the Materials and Methods section of the revised manuscript (lines 98-100).

Point 4: Which GFD compliance questionnaire was used? Was it a validated one? Please add a reference. And also add a reference for CDAT questionnaire.

Response 4: Thank you for this question. As regards the GFD compliance questionnaire, we used the validated tool developed by Biagi et al. (Biagi F, Andrealli A, Bianchi PI, Marchese A, Klersy C, Corazza GR. A gluten-free diet score to evaluate dietary compliance in patients with coeliac disease. Br J Nutr. 2009 Sep;102(6):882-7. doi: 10.1017/S0007114509301579). We also used a validated CDAT questionnaire proposed by Leffler et al. (Leffler DA, Dennis M, Edwards George JB, Jamma S, Magge S, Cook EF, Schuppan D, Kelly CP. A simple validated gluten-free diet adherence survey for adults with celiac disease. Clin Gastroenterol Hepatol. 2009 May;7(5):530-6, 536.e1-2. doi: 10.1016/j.cgh.2008.12.032). The relevant information was provided in the Materials and Methods (Dietary consultations) section of the revised manuscript (lines 132-135).

Point 5: How was GFD compliance assessed in children? An age-specific adapted questionnaire should be used.

Response 5: For pediatric patients, the questionnaires were filled by the parents or guardians, whereas older children actively participated in the process. This is a standard approach in research studies involving dietary recalls:

- Dowhaniuk et al. (2020), J Can Assoc Gastroenterol.: “The parents were asked to rank their child’s GFD adherence using a Likert scale from one to five.

- Czaja-Bulsa and Bulsa (2018), Nutrients.: “In the case of the youngest children (under seven years of age), dietary declarations were made by their guardians

- Garg and Gupta (2014), Int Sch Res Notices.: “Diet recall was done by parents for children in preschool age up to 5 years since parents were the only one giving the eatables to these children. Children, above 5 years of age, going to school and interacting with peers, were actively involved in the dietary recall along with the parents”.

The relevant information was provided in the Materials and Methods (Dietary consultations) section of the revised manuscript (lines 132-135).

Point 6: Don’t you think that unbalanced gender distribution could have affected your results? Please comment about this eventuality

Response 6: We would like to thank the Reviewer for pointing this out. According to European CD epidemiology data, females are much more likely to suffer from CD. Our study reflects the distribution of age and gender groups among the patients of Professor Antoni Gębala Children's University Hospital in Lublin, as well as among adults registered in the Polish Celiac Society in Lublin Voivodeship. To the best of our knowledge, gender distribution should not have affected our results.

Point 7: It would be more informative if symptoms in table 2 are presented spit for children and adults.

Response 7: Table 2 was modified.

Point 8: You reported that: “the transition to a GFD decreased the frequency and severity 161 of epileptic episodes in only two cases.”. Please, discuss this result: was cerebral MR/CT performed in these patients? Were there cerebral calcifications?

Response 8: The neurological symptoms reported before and after diagnosis were indicative of neuropathy. Two patients undergoing neurological treatment were diagnosed with less frequent epileptic seizures. The presented data were obtained from patient interviews or their medical history. Our study focused solely on dietary habits and the patients’ nutritional status.

Point 9: In most participants, the transition to a GFD improved intestinal health and contributed to the regeneration of intestinal villi, which reduced the frequency of and, subsequently, eliminated abdominal pain and diarrhea. Hematological parameters (hemoglobin, hematocrit, red blood cell counts, iron, ferritin) improved.” These sentences are not supported by numerical data.

Response 9: We fully agree with the Reviewer on this point. The above statement was deleted (lines 186-188). We analyzed the lifestyle choices and health indicators of CD patients, including blood biochemical parameters. The study will be continued in the future, and the results that were not presented in this manuscript (due to length constraints) will be published in our upcoming papers.

Point 10: What about BMI centiles (or z-scores) in children? Please add these data.

Response 10: We fully agree with the Reviewer on this point. BMI percentiles based on the WHO growth chart were added (lines:: 193-207).

Point 11: „These observations indicate that severe symptoms of CD and unresolved health issues can prompt patients to adhere to a GFD despite the encountered problems.” This is not a result and should be moved to the discussion section.

Response 11: We agree with the Reviewer on this point. This statement was moved to the Discussion section (lines 349-350).

Point 12: Some of the data commented on in the discussion are not reported in the results (i.e. only 10 patients diagnosed with CD consulted a dietician.

Response 12: We would like to thank the Reviewer for pointing this out. The relevant data were added in the Results section (lines 171-173).

Point 13: Is there a refunding policy for GFD products for celiac patients in your country? Please add this point, because it could affect GFD compliance and vary among different countries.

Response 13: We appreciate the Reviewer’s suggestion. The information about the Polish reimbursement policy concerning the purchase of GFD products by celiac patients was added (lines 66-71).

We are grateful to the Reviewer for constructive and insightful comments which have enabled us to substantially improve our manuscript. The manuscript was revised in line with the Reviewer’s suggestions, and we hope that it now merits publication in Nutrients.

Best regards,

Małgorzata Kostecka

Katarzyna Iłowiecka

Reviewer 2 Report

In this study, Kostecka et al aimed to evaluate and monitor the nutritional status of celiac disease (CD) patients, to explore the problems associated with diet planning and dietary adherence among children and adults, and assess the impact of these factors on the persistence of CD symptoms. At the start of the study and after 9 months, all adult participants were subjected to a body composition analysis with the SECA mBCA 515 analyzer.

They found that patients with gastrointestinal symptoms were significantly more likely to avoid even accidental exposure to gluten and were more likely to strictly follow GFD recommendations (p=0.0001) and safety guidelines when preparing meals at home (p=0.0023). Parents, in particular parents of toddlers and preschoolers, adhered strictly to dietary guidelines and did not allow for any exceptions when preparing meals (p=0.001). Persons at risk of malnutrition were also far less likely to deliberately choose gluten-containing foods (p=0.021), in particular patients with Marsh type 3a and 3b classification (p=0.01) and persons whose intestinal histology scores did not fully improve after switching to a GFD. An assessment of the effectiveness of diet therapy based on the phase angle revealed that dietary recommendations had a positive impact on patients who had been recently diagnosed with CD. In all age groups, the major problem was the accidental exposure to gluten, in particular in foods that were not labeled with the crossed grain symbol. The comparative analysis of CDAT questionnaires showed that dietary advice on eating out significantly improved adherence to a GFD and lowered the frequency of unintentional gluten exposure in all age groups.

They concluded that CD patients with a wide range of symptoms and a predominance of digestive symptoms tended to adhere more strictly to a GFD.

The study is of interest since a strict gluten-free diet is still the current and only proven treatment for CD patients. However, However, some issues deserve further elucidation and discussion.

Since strict adherence to GFD is mandatory to achieve symptom improvement, it is recommended by all current guidelines (Current guidelines for the management of celiac disease: A systematic review with comparative analysis. World J Gastroenterol. 2022 Jan 7;28(1):154-175) not only strict adherence but also regular follow-up maintenance possibly by providing the tools to ensure adherence such as support from patient associations. The authors should specify whether this recommendation is provided to patients after diagnosis because this could influence the degree of adherence to the diet. It would be more clear to modify the manuscript title to refer to the "evaluation of nutritional status and problems with dietary compliance in patients with celiac disease in Poland".

-The authors presented Comorbidities of their CD patients with a list of well-known autoimmune disorders that are associated with CD. Discussing the clinical impact of strict GFD, they should recall the importance of GFD in avoiding the secondary autoimmunity (Coeliac disease and secondary autoimmunity. Dig Liver Dis. 2002 Jan;34(1):13-5) as demonstrated by the occurrence of other immune- and autoimmune associated conditions in celiac disease possibly favored by gluten ingestion (Prevalence of silent coeliac disease in atopics. Dig Liver Dis. 2000;32:775-9; Anti-ganglioside antibodies in coeliac disease with neurological disorders. Dig Liver Dis. 2006 Mar;38(3):183-7).

-Regarding the finding that "CD patients with a wide range of symptoms and a predominance of digestive symptoms tended to adhere more strictly to a GFD" the authors should discuss a similar finding in a recent study published in Nutrients (Risk of Drop-Out from Follow-Up Evaluations for Celiac Disease: Is It Similar for All Patients? Nutrients. 2022 Mar 14;14(6):1223.).

Author Response

Responses to Reviewer #2

(changes in the manuscript are marked in blue)

Dear Reviewer,

Thank you for the positive feedback. Our point-by-point responses to your comments are presented below. The manuscript was revised in the “track changes” mode. We hope that the revised manuscript is suitable for publication in Nutrients.

The study is of interest since a strict gluten-free diet is still the current and only proven treatment for CD patients. However, some issues deserve further elucidation and discussion.

Point 1: Since strict adherence to GFD is mandatory to achieve symptom improvement, it is recommended by all current guidelines (Current guidelines for the management of celiac disease: A systematic review with comparative analysis. World J Gastroenterol. 2022 Jan 7;28(1):154-175) not only strict adherence but also regular follow-up maintenance possibly by providing the tools to ensure adherence such as support from patient associations. The authors should specify whether this recommendation is provided to patients after diagnosis because this could influence the degree of adherence to the diet. It would be more clear to modify the manuscript title to refer to the "evaluation of nutritional status and problems with dietary compliance in patients with celiac disease in Poland”.

Response 1: We appreciate the Reviewer’s suggestion. To address the Reviewer’s concerns, we have provided a general description of the support scheme offered by the Polish Coeliac Society (lines 72-79). The title of manuscript was also modified.

Point 2: The authors presented Comorbidities of their CD patients with a list of well-known autoimmune disorders that are associated with CD. Discussing the clinical impact of strict GFD, they should recall the importance of GFD in avoiding the secondary autoimmunity (Coeliac disease and secondary autoimmunity. Dig Liver Dis. 2002 Jan;34(1):13-5) as demonstrated by the occurrence of other immune- and autoimmune associated conditions in celiac disease possibly favored by gluten ingestion (Prevalence of silent coeliac disease in atopics. Dig Liver Dis. 2000;32:775-9; Anti-ganglioside antibodies in coeliac disease with neurological disorders. Dig Liver Dis. 2006 Mar;38(3):183-7)

Response 2: We would like to thank the Reviewer for pointing this out. We provided additional information about the clinical implications of GFD in patients diagnosed with diseases other than CD, as well as its impact on the development of secondary autoimmunity (lines 379- 386).

Point 3: Regarding the finding that "CD patients with a wide range of symptoms and a predominance of digestive symptoms tended to adhere more strictly to a GFD" the authors should discuss a similar finding in a recent study published in Nutrients (Risk of Drop-Out from Follow-Up Evaluations for Celiac Disease: Is It Similar for All Patients? Nutrients. 2022 Mar 14;14(6):1223.).

Response 3: This is a valuable suggestion. The indicated problem was analyzed in greater detail in the Discussion section (lines 345 - 360).

Once again, we would like to thank Reviewer 2 for constructive and insightful comments which have enabled us to substantially improve our manuscript. We hope that our results will be of interest to a wide range of readers dealing with the gluten-free diet in celiac disease. The manuscript was revised in line with the Reviewer’s suggestions, and we hope that it now merits publication in Nutrients.

Best regards,

Małgorzata Kostecka

Katarzyna Iłowiecka

Round 2

Reviewer 1 Report

Thank you for the improvements to the paper.

I still have some minor concerns:

Response 1:

what do you mean by “these procedures had been carried out several times in around 10% of the patients”?

Response 6:

I do know that females are much more likely to suffer from CD, but gender distribution is not so unbalanced in European epidemiological studies. Please comment on this point.

Response 8:

“The presented data were obtained from patient interviews or their medical history. Our study focused solely on dietary habits and the patients’ nutritional status.” This could represent a gap, please add this point among the limitations.

In table 2 there is only one p-value column, but it is not clear if it refers to the comparison before/after GFD in children or adults; please specify (and add one) accordingly.

Author Response

Responses to Reviewer #1 (round 2)

(changes in the manuscript are marked in orange)

Dear Reviewer,

We are grateful for your critical assessment which has enabled us to improve the quality of our manuscript. Our point-by-point responses to your comments are presented below. The manuscript was revised in the “track changes” mode. We hope that the revised manuscript is suitable for publication in Nutrients.

Point 1 - Response 1: what do you mean by “these procedures had been carried out several times in around 10% of the patients”?

Response: According to the medical history, two children and seven adults were diagnosed in more than one medical center, and gastroscopy was performed and biopsy samples were collected and graded based on Marsh classification several times. This information was provided to explain that all participants, including children, were always diagnosed with celiac disease based on gastroscopy results.

Point 2 -Response 6: I do know that females are much more likely to suffer from CD, but gender distribution is not so unbalanced in European epidemiological studies. Please comment on this point.

Response: Thank you for pointing this out. We agree that gender distribution is not so unbalanced in European epidemiological studies. However, please note that participation in the study was voluntary, and the participants were recruited by the Polish Celiac Society and by physicians/gastroenterologists of Professor Antoni Gębala Children's University Hospital in Lublin and the gastroenterology ward of the Independent Public Clinical Hospital No. 4 in Lublin. We had no influence on the small number of male participants. This could also be due to the fact that women are usually responsible for preparing meals at home, which is why they were more interested in the project. In our opinion, the uneven sex ratio had no effect on the results of the study.

Point 3 -Response 8: “The presented data were obtained from patient interviews or their medical history. Our study focused solely on dietary habits and the patients’ nutritional status.” This could represent a gap, please add this point among the limitations.

Response: We fully agree with the Reviewer. This point was added to the Strengths and limitations section (lines 414-415).

Point 4 - In table 2 there is only one p-value column, but it is not clear if it refers to the comparison before/after GFD in children or adults; please specify (and add one) accordingly.

Response: Table 2 was modified.

We are grateful to the Reviewer for constructive and insightful comments which have enabled us to substantially improve our manuscript. The manuscript was revised in line with the Reviewer’s suggestions, and we hope that it now merits publication in Nutrients.

Best regards,

Małgorzata Kostecka

Katarzyna Iłowiecka
